# Landscape Analysis of Public Health Jobs in India to Develop an Evidence-Based Public Health Curriculum

**DOI:** 10.3390/ijerph192315724

**Published:** 2022-11-25

**Authors:** Ashish Joshi, Ashruti Bhatt, Mahima Kaur, Ashoo Grover

**Affiliations:** 1School of Public Health, University of Memphis, Memphis, TN 38152, USA; 2Foundation of Healthcare Technologies Society, Delhi 110066, India; 3Indian Council of Medical Research, New Delhi 110029, India

**Keywords:** public health, public health competencies, public health education, public health workforce

## Abstract

The increase in communicable and non-communicable disease incidence and prevalence, changing population demographics, along with concerns about pandemics, natural disasters, and wars, have highlighted the challenges faced by health systems. The study aims to identify data on publicly posted public health jobs available to applicants eligible to work in India to identify the public health and allied fields workforce needs, skills, and expertise in India. A cross-sectional study was done in June–July 2021. The data was collected from eleven common job portals in India. Descriptive and content analysis was done to identify the most common job titles, educational level preferred/desired, skills, and experience required in the public health jobs in India. In total 382 unique public health and related fields jobs were analyzed. Job postings were most commonly classified as manager (*n* = 68), officer/lead (*n* = 61), analyst (*n* = 49), and consultant (*n* = 44). Around one-fifth of the jobs were based in Delhi (*n* = 98, 24%). About a quarter of the job postings required more than 8 years of experience (26%, *n* = 100). More than half of the job postings mentioned having the knowledge and understanding of data analysis and statistical approaches (*n* = 116, 64%). Around 15% (*n* = 193) of the job posting wanted the candidate to have expertise in communication. Skills were classified into various types such as software, technical, and language. Timely assessment of the curriculum should be done to impart skills related to the needs of the employers and prepare a skilled and competent public health workforce to address the 21st century public health challenges.

## 1. Introduction

Public health education is recently gaining momentum in India [1]. Historically in India, public health education was mainly delivered through medical schools but there has been a shift from medical schools to public health schools to develop a multi-professional public health specialist workforce [2].

The Joint Working Group UK-India embarked on the Master of Public Health (MPH) curriculum under the direction of the Joint Secretary (Medical Education and Training) and held its first meeting on 3 September 2014 in New Delhi and the second meeting in January 2015 took place in London, UK [3]. This led to the formation of a Taskforce on Public Health Education (PHE) in India under the Sub-group on Health Education and Training with expert members from the two countries (United Kingdom and India) in 2015 [3]. Following this, a detailed discussion between the experts from both countries resulted in the development of a curriculum guideline for MPH in India [3].

The number of Master of Public Health (MPH) programs in India has expanded rapidly with 44 public health programs in 2016–2017 from the first MPH program which was launched in the year 1995 [2]. Two main programs are currently being offered namely (i) core public health programs (Generic Masters in Public Health (MPH)) and (ii) specialized courses (MPH with tracks/specialization). Masters and diploma programs in health and hospital management/administration, epidemiology, health economics and healthcare financing, biostatistics and data management, public health nutrition, are some of the specialized courses in public health offered at some institutions in India [3,4].

Nowadays, in addition to some governmental jobs, opportunities are slowly emerging for MPH graduates in the private sector, and in sectors such as development, pharmaceuticals, and information technology. Additionally, MPH graduates possessing strong analytical skills are involved in work related to healthcare-related “big data” initiatives [2].

However, despite this growth, public health education in India is facing challenges related to accreditation of institutions and programs, competency-driven curriculum, and career pathways for trained public health professionals [4]. There is a need for identifying the gaps in the existing academic programs as an adequately trained multi-professional workforce can effectively address public health challenges of the 21st century and hence contribute to reducing the burden of diseases [5]. In 2010, the Lancet Commission for Education of health professionals for the 21st century, recommended India revisit and transform its existing institutional and instructional frameworks of public health education to meet and respond to emerging multifaceted present-day challenges [4,5].

The extent to which the public sector health services can be improved, depends largely upon the preparedness and skillfulness of the public health workforce along with other dimensions of health system, which is in turn dependent upon the quality of its education and training [6]. The Bhore’s Health Survey and Development Committee in 1946, emphasized—the inadequate teaching of preventive medicine and public health in the medical training, which highlighted the need and importance of mainstreaming the public health education in healthcare sector [7]. The development of a skilled public health workforce is an essential prerequisite for finding solutions to enormous public health problems and challenges in the country. There is a need to put efforts in defining the core competencies required to get into the public health system. However, some of the essentials for public health professionals (doctors, nursing staff, paramedical workers, grass-root workers, allied health workers along with other scientific areas such as biomedicine, sociology, pedagogy, care sciences, anthropology, etc.) [8,9] are knowledge, attitudes, skills, and abilities, that they should possess to effectively deliver essential public health domains such as epidemiological surveillance, situation assessments, and health promotion [10]. The gap between the competencies and specialized areas of public health needs to be bridged by segregating the competencies into core and functional ones. The core competencies will be a collective of crosscutting skills whereas functional competencies should be discipline based [11].

Additionally, there is a challenge of the knowledge and skills acquired during the program vs. what is demanded by the job [12]. In order to meet the ever-changing needs of the public health system in terms of the requisite knowledge, skills, attributes, leadership and expertise, it is important to ascertain job specifications.

In India, the MPH course is designed to be a two years full-time program including internship and dissertation. The two months internship is undertaken by all the public health candidates with an aim to integrate learning and practice in an active public health organization. The course comprises 15 core modules and several elective modules which may be offered by universities depending on their capacity and capability. The four elective streams include (a) Epidemiology, (b) Health system management, (c) Health Programme, Policy, and Planning, and (d) RMNCH + A (Reproductive, Maternal, Newborn, Child and Adolescent Health) [3]. Table 1 gives in detail the Skills, Values, and Competencies mentioned in the MPH curriculum expected to be demonstrated/developed by students [3].

Therefore, there is a need to have a clear understanding of whether the present training programs are effectively preparing students for multi-professional fields (including medical, non-medical and allied health professionals) to enter the field of public health with a wide range of skills and competencies. The objective of the study is to (a) Identify data on publically posted public health jobs available to applicants eligible to work in India to ascertain what the public health and allied fields workforce needs, skills, and expertise in India are and (b) To analyze the unique public health jobs and categorize them as per various job positions (c) To analyze educational competencies required by available vacancies on common job portals across India. Our study will be the first study in the country to evaluate the skills, knowledge, competencies, abilities, behavior attributes and expertise that are required by the public health professionals in order to be employable in an Indian setting. It further provides evidence for universities and colleges offering generic and specific disciplines MPH degrees to review their curriculum to ensure that MPH graduates are equipped with abilities and skills required by the current public health employers.

## 2. Materials and Methods

For this cross-sectional study, data was collected from ten common job portals in India. Descriptive and content analysis was done to identify the most common job titles, degree level preferred/desired, skills, and experience required in public health jobs using keyword searches for work in India. The extraction of the qualitative data and compilation of the information available was done with a thorough understanding and analysis of the provided information by identifying synonymous concepts. The study follows the equator observational study checklist for reporting the study (Appendix A) [13].

### 2.1. Search Period

The data was cumulatively collected from June to July 2021 across eleven different job portal including Ambition box, Naukri, Freshers world, Jooble, Glassdoor, Monster India, Indeed, LinkedIn, Times job, DevNet and NgoBox.

### 2.2. Inclusion and Exclusion Criteria

While searching for the jobs we applied three filters (a) quotation marks were used to specify each keyword, (b) Location was set to be “India”, (c) only jobs posted in the last one week of the search date was included in the study and (d) if the total number of jobs retrieved were more than 100 then only top 100 job postings were included in the study. NgoBox did not show any job for the specific keywords, but since it was a common job portal for public health-related fields jobs only the filter of jobs in past week was applied. Unique jobs within and across the searched job portals were analyzed. The job postings were excluded from the analysis if they (a) Did not mention an educational qualification, (b) job not related to the selected keywords, (c) requires a nursing certificate with no PH or allied fields education (d) requires an MBBS/BDS/Pharmacist certificate with no PH or allied fields education, (e) International job posting.

#### Definition of Public Health Jobs

Rotem et al. defined public health practitioners as people who were “engaged in activities related to the protection (promotion and/or restoring) of the collective health of whole or specific populations (as distinct from activities directed to the care of individuals)” [14]. Jobs specific to the role of public health practitioners including doctors, nurses, allied health professionals such as physiotherapist, psychologist, health informatics etc [15] were included.

### 2.3. Data Selection

In the current study the job postings were reviewed by two reviewers (AB & MK) screening independently the jobs present across each job portal using the defined keywords. Any disagreements between both reviewers were resolved through consulting a third reviewer (AJ).

### 2.4. Data Extraction and Analysis

Following variables were extracted from each job posting and included:Job Title: Data was recorded from the publicly available jobs across various job portals in India. These jobs were further characterized into different categories based on the job title and the job description. The emerged major relevancies were noted and classified according to the meaning they conveyed. Further, the qualitative data collected was coded in order to identify patterns across the data in relation to the title and description. For example:
Analyst: Job title was classified as an analyst if its main function was to evaluate different health programs, focusing on the analysis of large-scale data reporting projects. Biostatistician, Data analyst, Biostatistician manager, and statistician in the title were classified as analysts.Director: Program Director, Academic Director, Deputy Chief of Party, and director word specified in the job title were classified under director. The Director’s role included performing administrative work and developing a team, building and designing programs, services, and initiatives. Responsible for maintaining and improving the efficiency and effectiveness of all areas under his/her direction and control.Faculty: Job postings by educational institutes/universities for teaching the students were classified under the faculty category.Researcher: The job posting requiring the candidate to devise, formulate, and execute study protocols and disseminate their insights through the publishing of findings were classified as a researcher.Manager: Project Manager, Program Officer/Lead, Policy Manager, states operations manager in the title along with the responsibility of project/program management financial, training, strategic planning, operations, or personnel management were classified as Mangers.Consultant: Those job postings that demanded candidates to provide expert opinions, analysis, recommendations, and providing strategies to prevent problems and improve performance to organizations or individuals were classified as consultants.
Organization type: The name and types of the organizations of the available jobs postings were recorded. They were then categorized into (a) Non-Profit Organization, (b) Consulting Organization, (c) Pharmaceutical organization, (d) Contract Research Organization (e) Academia, (f) International Health agency, (g) Government of India (h) Health and Hospital and (i) OthersLocation: The location of the vacancies across different Indian states was also documented.Educational Qualification (Required/Preferred): Data were recorded for both required and preferred qualification requirements mentioned in the job posting. Information recorded on the MPH and the allied fields included public health, epidemiology, biostatistics, informatics, nutrition. They were then classified into the following: (a) Bachelors (any) and above, (b) Masters (any) and above, (c) Ph.D. and above, (d) Medicine (Doctorate) plus Masters.Work experience: Information was extracted on work experience needed, i.e., required or preferred to apply for the various jobs available.Salary: Information on the available salary information gathered across the various job postings.Job type: Job postings were screened to record data on the job type including contract, full-time, internship, part-time or temporary.Knowledge, competencies, and expertise: Information gathered regarding the knowledge, competencies, and expertise mentioned for the available public health jobs.Skills: Information was also recorded to gather information on the type of skills. Descriptive analysis was also performed to determine the most common software skills listed with each job description.

### 2.5. Statistical Analysis

Descriptive analysis was performed to report mean and frequency distribution for the continuous and the categorical variables. Results for various job characteristics such as job title, organization, salary requirements, educational qualification, work experience, and skills identified were presented as percentages. Cross tabulations were conducted to examine the degrees required/preferred and experience required/preferred across each of the job categories. Cross-tabulations were conducted to examine the distribution of skills across the various job categories. SPSS v.24 and Microsoft Excel v.2016 were used to analyze the data.

## 3. Results

### 3.1. Identification of Jobs across Job Portals

Common Indian job portals were searched to retrieve the job related to the field of public health to understand skill sets requirements. Vacancies were retrieved from eleven job portals including Ambition box, Naukri, Freshers world, Jooble, Glassdoor, Monster India, Indeed, LinkedIn, Times job, DevNet and NgoBox. Filters were applied and the retrieved jobs URL and job posting was saved to a spreadsheet and word document, respectively. Job postings that were identical within the different keywords across the same job portal or other included job portals were removed from the study as duplicates. After eliminating a total of 678 duplicate vacancies, 1055 vacancies were excluded based on the exclusion criteria. In total 382 unique public health and related fields jobs were analyzed (Figure 1).

### 3.2. General Characteristics of the Job Postings

Results of our search showed an overall 382 number of jobs across the most common job search engines (Table 2) for public health jobs in an Indian setting.

### 3.3. General Characteristics of the Job Postings

The most common job titles were characterized as manager (*n* = 68, 18%), officer/lead (*n* = 61, 16%), analyst (*n* = 49, 13%), and consultant (*n* = 44, 12%). These were followed by other categories such as coordinator, researcher, and academic faculty (Figure 2). More than one-third of the hiring organizations for these jobs were non-profit organizations (37%, *n* = 132) followed by consulting organizations (16%, *n* = 56), a pharmaceutical company (12%, *n* = 44) and academia (11%, *n* = 38). Around one-fifth of the jobs were based in Delhi (*n* = 98, 24%). About 17% (*n* = 69) of the mentioned jobs were based in Karnataka followed by 12% (*n* = 49) in Maharashtra (Table 3).

About 47% of the jobs required an educational level of master’s or above while more than half of the job postings (61%) preferred the candidates to have an educational qualification of master’s or above (Figure 3a). The distribution of required education qualification that mentioned masters and above as the most essential are manager (65%, *n* = 44), analyst 63% (*n* = 31), consultant 57% (*n* = 25), researcher 48% (10), coordinator 46% (*n* = 12) and office/lead 44% (*n* = 27) (Figure 3b). Additionally, the preferred education qualification across the most common job categories was also masters or higher for each of the following job categories: consultant 76% (*n* = 13), manager (68%, *n* = 13), office/lead 68% (*n* = 17), director 64% (*n* = 7), analyst 61% (*n* = 11) and coordinator 38% (*n* = 5) (Figure 3c). Length of required experience was mentioned in 84% of the jobs. About a quarter of the job postings required more than 8 years of experience (26%, *n* = 100) followed by 5 to 7 years of experience in 29% of the job postings (*n* = 111). Almost 95%(*n* = 363) of the included jobs were full-time. More than half of the job postings did not mention the salary (86%, *n* = 327) (Table 3).

### 3.4. Knowledge, Competencies, Behavioural Attributes and Expertise

Table 4 outlines the knowledge, competencies, and expertise listed in the included job postings. Knowledge and understanding of IT applications were mentioned in over one third of the jobs (*n* =134, 34%), followed by medical and clinical knowledge (*n* = 31, 8%) and management and planning (*n* = 31, 8%). The knowledge and understanding of public health allied fields topics are mentioned in Table 4 with knowledge and understanding of general public health (*n* = 29, 7%) followed by knowledge of health system & government organizations, policies, schemes and programs, and governance (*n* = 17, 4%). More than half of the job postings mentioned having the knowledge and understanding of data analysis and statistical approaches (*n* = 116, 64%) followed by clinical research and trials (*n* = 42, 23%). Analysis of the behavioral attributes mentioned in the jobs stated that about 15% (*n* = 193) of the job posting wanted the candidate to have effective communication followed by team-worker (10%, *n* = 127) and ability to have strong relations with stakeholders and individuals (10%, *n* = 124).

### 3.5. Types of Skills Mentioned in Different Job Postings

Skills were classified into various types of skills such as statistical software, technical, language, and communication skills. The most commonly cited statistical programming language and statistical programs included Statistical Analysis System (SAS) (15%, *n* = 28), Structured query language (SQL) (13%, *n* = 23), Python (12%, *n* = 22), R language (15%, *n* = 25) and Statistical Package for the Social Sciences (SPSS) (11%, *n* = 21). The skills of the two main data visualization tools mentioned in the various job postings were Tableau and Microsoft Power BI (*n* = 10, 5%). Some of the main technical skills mentioned in the job postings were MS Office (70%, *n* =133), experience with any of the software related to data analysis and data visualizations tool (20%, *n* = 37), and skill and experience to work with Electronic Health Records (EHR) (2%, *n* =4) (Table 5). Figure 4 gives in detail the distribution of various software skills required/preferred stratified by job categories.

Almost half of the job postings mentioned that candidates should have good spoken (45%, *n* = 237) and written (44%, *n* = 236) communication skills. Presentation skills were mentioned in less than 10% of the job postings (*n* = 54). Fluency in the spoken English language was mentioned in more than half of the job postings (54%, *n* = 116), English literacy in about a fifth of the job posting (19%, *n* = 40), followed by regional languages spoken in India (19%, *n* = 40). Other skills mentioned in about a quarter job postings included interpersonal skills/collaboration skills (22%, *n* = 118), management skills (28%, *n* = 149), and analytical skills (26%, *n* = 138). A small percentage of job postings mentioned scientific writing (%, *n* = 25) and technical writing skills (12%, *n* = 63) (Table 5).

## 4. Discussion

The objective of the study was to identify data on publicly posted public health jobs available to applicants eligible to work in India and to identify the public health and allied fields workforce needs, skills, and expertise. The recent increase in the rate of non-communicable diseases such as hypertension, diabetes, and cancer, along with concerns about pandemics have highlighted the overburdened health systems [16,17]. Therefore, the creation of a dedicated public health workforce has been identified as one of the important prerequisites in mitigating the challenges faced by the health systems.

Results of our study showed that manager (*n* = 68), officer/lead (*n* = 61), analyst (*n* = 49), and consultant (*n* = 44) were the most common job categories offered. This also corroborates with the previous study by Cole et al. (2011) where leadership and management along with other essential public health skills were considered necessary in order to respond to the present and upcoming public health challenges. Moreover, it was also highlighted in the literature that greater fluency in cost-effectiveness analyses, strong political skills including leadership and advocacy, the ability to work collaboratively across diverse regions and sectors will help in tackling issues related to pandemics in future [18], this is also consistent with the WHO roadmap for developing the public health workforce, which emphasizes on specific competencies required, identification of essential domains and move towards a cross-disciplinary response to population health challenges [19].

The most common type of hiring organizations were non-profit organizations (37%, *n* = 132) followed by consulting organization (16%, *n* = 56). Surprisingly, in recent times NGOs, consulting firms, research organizations, and academic institutes have started hiring public health professionals rather than just governmental agencies. The job postings analyzed largely required an educational level of master’s or above. Additionally, people with background from diverse field such as social work, law, finance, human resources, management, anthropology, communication, statistics, and economics could have job opportunities in public health. Our study revealed that there are few jobs for candidates with less than 2 years of experience, which suggests that employers want candidates ready for work. Consequently collaborations and partnerships between public and private sectors need developing to create opportunities that develop students’ work skills, for example, an internship/dissertation where students can hone the skills identified to be critical in high demand. Such experience gained during the education at the grass-root level, funded research projects, and real-time data will strengthen students’ portfolios and they could integrate the knowledge, values, competencies, and skills they have attained in the classrooms.

The most common statistical programs and programing language critical in the job analysis were MS Office, SAS, SPSS, R, Python, MS SQL, STATA, and Tableau. Interestingly, the curriculum of MPH implemented in India focuses on only either SPSS or STATA to be taught to students as statistical software for data analysis. Additionally, knowledge and understanding of the IT applications were mentioned in more than quarter of the jobs (*n* =134, 34) and more than half of the job postings mentioned having the knowledge and understanding of data analysis and statistical approaches (*n* = 116, 64%). To be fit for purpose, the recent digital revolution demands graduates’ be up skilled in quantitative skills, data literacy, and analytical competence. In recent times the use of big data has been increased to tackle complex problems. Computers, the internet, and information technology have opened pathways to access, analyze, and interpret data to solve public health issues [20,21]. In health sciences and allied fields, having a STEM foundation (supports improvements in Science, Technology, Engineering and Mathematics education) knowledge, possessing sound technical skills, and having the ability to handle large amounts of data is a strength. Therefore, new graduates’ should be equipped with digital abilities and develop a working knowledge of some compiled and interpreted programming languages in data management, data mining, and visualization [22].

About half of the job positions analyzed mentioned that candidates should have good spoken and written communication skills along with good presentation skills. Other skills mentioned in about quarter of job postings included interpersonal skills/collaboration skills, management skills, and analytical skills. The positions were more competency-driven involving a participatory approach and ability to multitask and self-learning rather than stepwise supervisory functions. Therefore, in addition to inculcating more quantitative training, all MPH program graduates should be taught a curriculum that includes qualitative and behavioral aspects including effective communications, stakeholder engagement, data quality challenges, model interpretability, research ethics, privacy and security, health policy and regulatory guidelines [22]. Moreover, the CDC framework for essential public health services should also be taken in to cognizance to inculcate the required skills among public health professionals for overall health of the communities everywhere [23].

Results of this study indicates directions to be considered for the development of the curriculum by the various academic departments focusing on public health. It can also guide the evaluation of academic programs on an ongoing basis to identify skills that would be essential for students to acquire and make them job-ready in this rapidly changing environment. Furthermore, to develop the best-tailored curriculum, feedback from students should be obtained. Faculty members who work in academic institutions, research departments, and programs are expected to follow up on the changing requirements and update the content of their curriculum continuously [24]. During the COVID-19 pandemic, we saw existing inequalities, inequities within and among the countries all over the world, and the struggling health systems. An editorial published by Ghaffar et al. (2021) proposed at least four more areas for consideration by schools of public health in preparation for a post-COVID world. These included (a) training and expertise in supply chain management, (b) expertise and proficiency to identify and diminish the effect and spread of misinformation and fake news, (c) development of expertise in learning and application of technologies in the collection of data, synthesis of available information and dissemination of decisions promptly and (d) priority setting and resource allocation [25].

It was interesting to see that lot of jobs had to be excluded as they were unrelated to the selected keywords. Graduates should be made aware of the portals from where they can find jobs. Therefore, this study suggests that there is a need for a dedicated job portal focusing on public health and allied field jobs. The creation of a dashboard focusing on the demand and supply of the public health jobs, types of skills required/preferred, funding of public health research projects, and top public health employers would help to increase the scope of public health, job market, and employability rate. The portal should also be useful to the employers where they have easy access to profiles of public health professionals to be hired and post the job postings [26].

A few limitations of the study include that we might have not included public health job postings not be publicly posted (for example government jobs). Few jobs can be filled through networks. Additionally, we might have left out job postings posted on subscription-restricted job portals. We also limited ourselves to the top 100 job postings. Additionally, over a limited time there might be inclusion or exclusion of heightened demands for certain types of jobs.

## 5. Conclusions

To address the unmet needs in the field of public health, colleges and universities providing public health and allied programs should continuously be updating curricula to incorporate key public health functions particularly in the area of data science such as knowledge and understanding of data analysis and statistical approaches with desirable skills and competencies like skilling quantitative skills, data literacy, and analytical competence. A timely assessment of the curriculum should be done to impart skills related to the needs of the employers and prepare a competitive public health workforce. This study will help create a curriculum to prepare a competent cadre of professionals who have a basic understanding of the various aspects of public health and have the skills and knowledge to meet the population health challenges in the Indian context and perform in complex, intersectoral, and technology-oriented environments.

## Figures and Tables

**Figure 1 ijerph-19-15724-f001:**
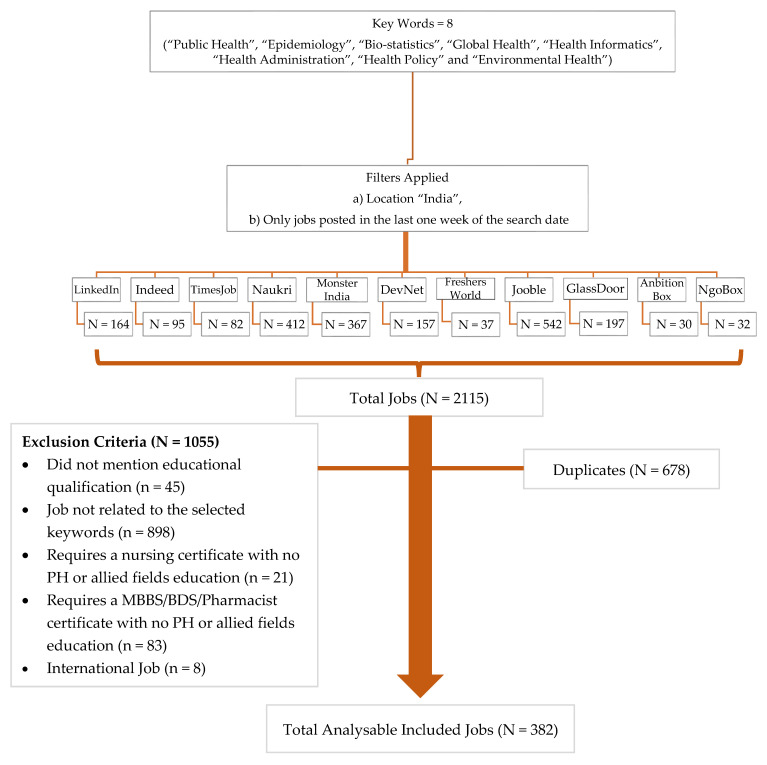
Methodological framework for search.

**Figure 2 ijerph-19-15724-f002:**
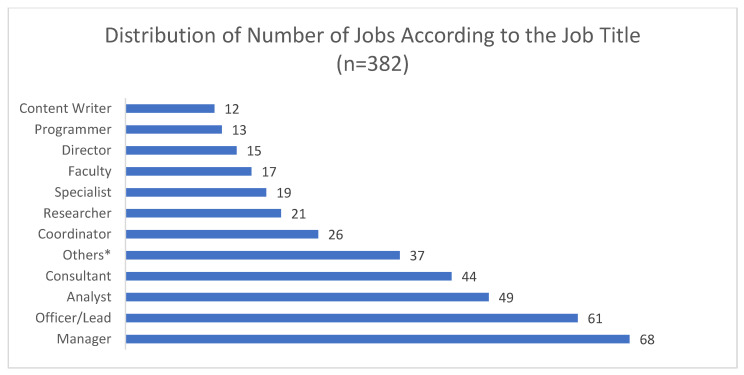
Frequency of job titles in included job portals. Others * = Advisor (*n* = 9), Associate (*n* = 9), Community worker (*n* = 2), Fellowship/Internship (*n* = 6), Healthcare Administrator (*n* = 1), Nutritionist (*n* = 3), Software designer or developer (*n* = 5), Technician (*n* = 2).

**Figure 3 ijerph-19-15724-f003:**
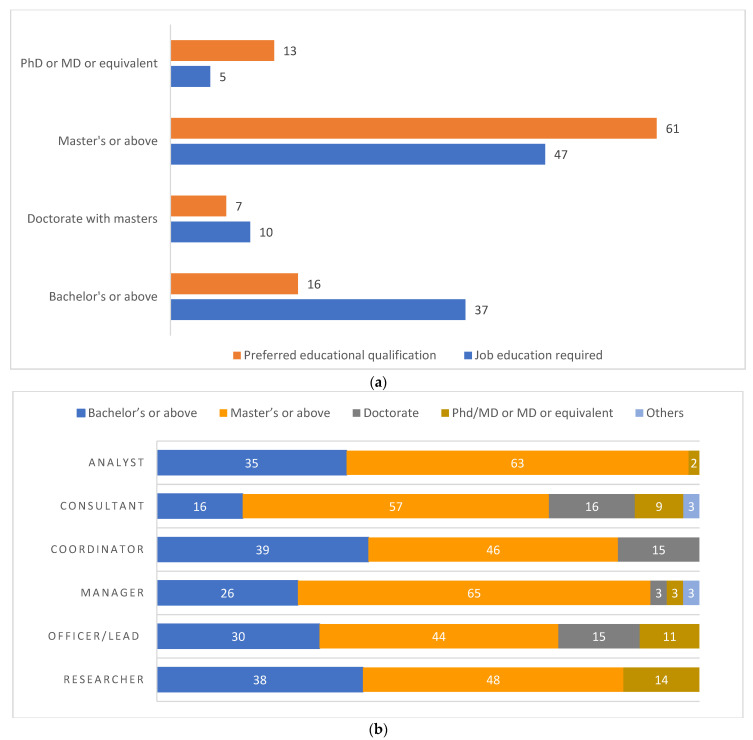
(**a**) Percentage distribution of the educational qualification (essential/required) in the various jobs. (**b**) Percentage distribution of the educational qualification required across the top six Job categories. (**c**) Percentage distribution of the educational qualification preferred across the top six Job categories.

**Figure 4 ijerph-19-15724-f004:**
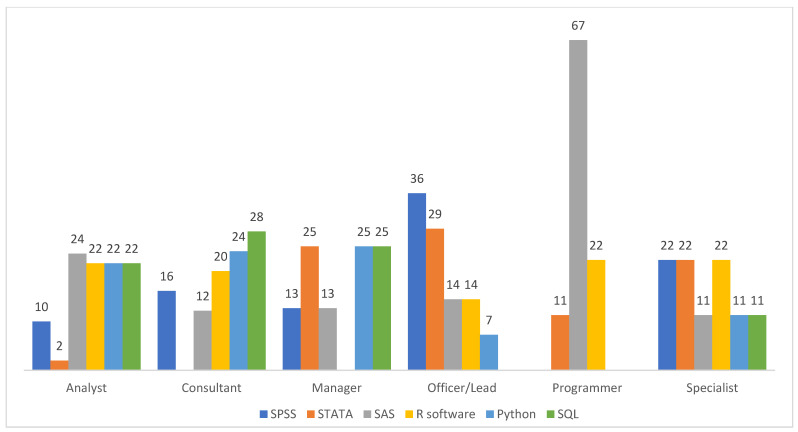
Percentage distribution of various software skills required/preferred stratified by top six job categories.

**Table 1 ijerph-19-15724-t001:** Skills, values, and competencies mentioned in the MPH curriculum expected to be acquired by students (Adopted from Ministry of Health and Family Welfare. 2017–2018 [3]).

Skills/Values/Competency	Skills, Values, and Competencies Mentioned in the MPH Curriculum Are Expected to Be Demonstrated/Developed by the Students.
Skills	Analytical and assessment skills for collecting and interpreting information Policy planning and development skills to address public health challenges Communication skills for advocacy, dissemination, and evaluation of public health data and information Financial planning and management skills for running public health programs in the country Leadership skills Undertake operational research in institutional and field settings. Demonstrate qualities of leadership and mentorship Apply contemporary ideas to influence program organization and management, problem-solving and critical thinking in the public health domain.
Values	Work in socially, culturally, and economically diverse populations by being attentive to the needs of vulnerable and disadvantaged groups and being well versed with existing health systems. Be an effective member of a multidisciplinary health team. Demonstrate ethics and accountability at all levels (professional, personal and social). Practice professional excellence, scientific attitude, and scholarship. Demonstrate accountability and responsibility Be open to lifelong learning
Competencies	
Competency 1: Apply the course learning to the public health system and its challenges:	Demonstrate adequate knowledge and skills to a wide range of public health topics Critically conduct the situational analyses and develop an action plan for identified public health issues Develop the workforce for tackling public health-related responsibilities in defined geographical areas Develop an understanding of the epidemiological transitions of populations specific to each State within the country to prioritize public health challenges for policymaking
Competency 2: Develop, implement and evaluate key public health policies:	Apply the conceptual framework to understand policy processes in health care Understand roles of supply, demand and absorption for public health personnel Facilitate inter-sectoral coordination and public-private partnerships Critically analyze resource allocation for competing for public health interests across programs Formulate context-appropriate policies and design programs to address public health challenges, effectively
Competency 3: Develop and demonstrate competency in managing health systems at different levels:	Identify short-term and long-term health program goals at national, state, and district levels Prioritize health issues in the population (situational and community diagnoses) Describe various managerial information systems and their applications Describe program management plans in health Apply core management principles for human resources in health Understand and apply program budgeting and economic interpretation Understand and apply quality assurance and improvement techniques in health care
Competency 4: Develop competency in research:	Understand and apply ethical principles in research, evaluation, and dissemination Develop competence to critically evaluate and appraise existing information and identify gaps to apply evidence based interventions to Indian settings. Formulate and test research hypotheses in real-world scenarios from existing data Translate research knowledge for evidence-based policymaking

**Table 2 ijerph-19-15724-t002:** Overview of the number of jobs resulting from each keyword across the most common job search engines.

Job Portal	Date of Search (Website)	Biostatistics	Environmental Health	Epidemiology	Global Health	Health Administration	Health Informatics	Health Policy	Public Health	Total Number of Jobs
Portal 1	25 June 2021	2	0	1	0	0	0	0	5	8
Portal 2	26 June 2021	0	0	0	0	2	0	0	4	6
Portal 3	21 June 2021	6	5	1	3	1	5	1	17	39
Portal 4	14 June 2021	3	3	0	3	1	1	0	19	30
Portal 5	26 June 2021	15	6	16	8	6	9	6	9	75
Portal 6	15 June 2021	16	2	8	4	1	1	0	25	57
Portal 7	21 June 2021	2	2	1	0	0	4	0	1	10
Portal 8	25 June 2021	5	1	3	7	0	0	0	13	29
Portal 9	23 June 2021	8	1	7	0	1	0	0	9	26
Portal 10	28 June 2021	1	1	6	13	1	0	3	69	94
Portal 11	7 July 2021	8								8

**Table 3 ijerph-19-15724-t003:** Descriptive analysis of the Location, Hiring organization type and salary across various Public health jobs.

Parameter	Attributes	*n* (%)
Location	Delhi	98 (24)
	Karnataka	69 (17)
	Maharashtra	49 (12)
	Telangana	26 (6)
	Haryana	25 (6)
	Tamil Nadu	24 (6)
	Uttar Pradesh	20 (5)
Hiring Organization Type	Non Profit	132 (37)
	Consulting	56 (16)
	Pharma	44 (12)
	Academia	38 (11)
	Contract Research Organization (CRO)	34 (9)
	International Health agency	23 (6)
	Government of India	16 (4)
	Health and Hospital	10 (3)
	Others ^1^	6 (2)
Salary	Not Mentioned	327 (86)
	Mentioned	61 (16)
	Volunteer	1 (0)

^1^ Others: Manufacturing (*n* = 1), Marketing (*n* = 1), Software (*n* = 1), Start up (*n* = 1), National Healthcare company (*n* = 2).

**Table 4 ijerph-19-15724-t004:** Knowledge, competencies, abilities and behaviour attributes across various job postings.

Knowledge/Competencies and Expertise		*n* (%)
A. Knowledge and Understanding of organisation	IT applications	134 (34)
	Others ^1^	49 (13)
	Medical and Clinical	31 (8)
	Management and planning	31 (8)
	Pharmaceutical (drug development and products, regulatory affairs)	14 (4)
	Health Informatics	13 (3)
	Understanding of organisational processes	12 (3)
A1. Knowledge and skillset of Public Health Allied Fields	Others ^2^	54 (86)
	General Public health	29 (7)
	Health system & Gov. organisations, policies, schemes and programs, and governance	17 (4)
	MNCAH & Family Planning	12 (3)
B. Understanding of Research	Data analysis and Statistical approaches	116 (64)
	Clinical Research and Trials	42 (23)
	Research principles and guidelines	23 (13)
C. Behavioural/Personal Attributes	Effective Communication and influence	193 (15)
	Team-worker (multicultural & multidisciplinary team)	127 (10)
	Strong relations with stakeholders and individuals	124 (10)
	Strategic, creative, attention to detail, and critical thinker	102 (8)
	Self-starter and independent worker (self-motivated)	99 (8)
	Adaptability (meeting deadlines, prioritizing)	97 (8)
	Problem Solver	77 (6)
	Leadership (entrepreneur mind-set)	70 (5)
	Multi-tasker	65 (5)
	Interpret and produce results (present data and reports)	62 (5)
	Organiser	47 (4)
	Decision maker	45 (4)
	Proactive (propose new ideas & initiatives, meeting objectives, performance oriented)	40 (3)
	Training and institutional capacity building	32 (2)
	Innovation (innovative mind-set) & Organisational learning	30 (2)
	Others ^3^	27 (2)
	Work ethics (accountability, empathy, confidentiality)	25 (2)
	Creating empowering and motivating environment	19 (1)

^1^ Others: Monitoring and evaluation principles (*n* = 3), Finance and accounting (procurement processes) (*n* = 10), Marketing (*n* = 10), Healthcare industry (*n* = 1), Behaviour change communication (*n* = 1), Environment health & safety (*n* = 7), HEOR (*n* = 2), Data collection (quality and accuracy) (*n* = 7), Data management (*n* = 8). ^2^ Others: Global health & development (*n* = 1), COVID-19 response (*n* = 3), Infectious diseases (*n* = 4), Epidemiology (*n* = 10), Community health engagement (*n* = 1), Immunization context (*n* = 2), Nutrition (*n* = 6), WASH issues (*n* = 3), Humanitarian (*n* = 8), Policy development (*n* = 10), Mental Health issues in adolescents and children (*n* = 2), Tuberculosis (*n* = 2), HIV-AIDS (prevention, care, support, and treatment) (*n* = 2). ^3^ Others: Forward-thinker (*n* = 3), Respect and promote individual and cultural differences (*n* = 7), Change in settings and community (*n* = 6), Negotiation (*n* = 9).

**Table 5 ijerph-19-15724-t005:** Type of Skills required by different job postings related to public health and allied fields.

Job Category	Skills	*n* (%)
Software skills	Others ^1^	53 (28)
	SAS	28 (15)
	R software	25 (14)
	SQL	23 (13)
	Python	22 (12)
	SPSS	21 (11)
	STATA	12 (7)
Communication skills	Spoken	237 (45)
	Writing	236 (44)
	Presentation Skills/Creative skills	54 (10)
	Others ^2^	4 (1)
Technical skills	MS Office	133 (70)
	Data Anaysis Software& data visulization tools	37 (20)
	Cloud system (AVS)	11 (6)
	Others ^3^	8 (4)
Others skills	Management Skills	149 (28)
	Analytical Skills	138 (26)
	Interpersonal Skills/Collaboration skills	118 (22)
	Technical Writing Skills	63 (12)
	Scientific Writing	25 (5)
	Clinical skills	20 (4)
	Others ^4^	19 (3)
Language	English	116 (54)
	Hindi Language	57 (26)
	Regional Language	40 (19)
	Others ^5^	3 (1)

^1^ Others: Tableau (*n* = 10), Power BI (*n* = 9), S plus (*n* = 1), StatXact (*n* = 1), Sharepoint (*n* = 1), SAP Business Objects BI (SAP BO) (*n* = 1), IMS proprietary databases (*n* = 1), VBA/ macros (*n* = 3), ADaM (The Analysis Data Model) (*n* = 2), Study Data Tabulation Model (SDTM) (*n* = 2), Microsoft Access (*n* = 4), Alteryx (*n* = 1), Infogram (*n* = 1), Epi-info (*n* = 6), CSPro (*n* = 1), N-vivo (*n* = 2), Atlas (*n* = 1), Epicollect (*n* = 1), DHIS (*n* = 1), MATLAB (*n* = 3), Metabase (*n* = 1). ^2^ Others: Listening skills (*n* = 4). ^3^ Others: Multimedia (*n* = 3), IQVIA Global Data Model (GDM) (*n* = 1), Electronic Health records (EHR) (*n* = 4). ^4^ Others: Marketing skills (*n* = 7), Quality control/content quality control (*n* = 6), Digital Technology skills (*n* = 1), Creative/Innovative skills (*n* = 5). ^5^ Others: Foreign language (*n* = 3).

## Data Availability

The data presented in this study are available on request from the corresponding author.

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
