# Peer review of "Landscape Analysis of Public Health Jobs in India to Develop an Evidence-Based Public Health Curriculum"

_ijerph, 2022, doi:10.3390/ijerph192315724_

Round 1

Reviewer 1 Report (Previous Reviewer 2)

Dear Colleagues,

I have re-reviewed this paper, which is much improved. There are some stylistic issues that I have indicated in track changes in a Word version of the article.  There are a few other substantive issues that need addressing. 

Author Response

Reviewer 2 Report (New Reviewer)

Please find my comments in the attached document.

Round 2

Reviewer 2 Report (New Reviewer)

Dear authors,

thank you for your work on this article. I appreciate your changes and adaptations. The one thing I continue to struggle with is your objective c) "to compare educational competencies acquired with available vacancies on common job portals in India."

I don't see where in the manuscript you are comparing educational competencies that were acquired (by whom?) with what is asked in the job ads (I imagine that is what you meant to do?). I see the section 3.4 on Knowledge, competencies, behavioural attributes and expertise but I can't connect it with the objective.

Can you either clarify your objective or add information in the results section on this comparison?

Thank you very much again.

Author Response

This manuscript is a resubmission of an earlier submission. The following is a list of the peer review reports and author responses from that submission.

Round 1

Reviewer 1 Report

Please refer to the attached pdf for my recommendation and feedback for the authors.

Author Response

Thank you for reviewing the manuscript and providing your valuable comments and suggestions. All the minor corrections suggested has also been corrected in the updated version.

Reviewer 2 Report

Dear Colleagues,

Many thanks for sending me this study to review. It  reviews what employers want from people they wish to recruit for public health related jobs and outlines the competencies that are agreed on for MPH graduates in India. It however, does not compare the two directly and this is a major weakness.

In addition it assumes that what is asked for by employers directly translates into what the degree should be train people in.  What is required by the country's health system towards Universal Health coverage needs to feature more prominently in the commentary and discussion.

The term Public Health is used loosely and interchangeably in the introductory paragraphs. On the one hand it seems to be the same as public sector health services, and the authors remark that doctors, nurses, paramedics are public health workers. In fact the researchers are looking more at public health as positions and competencies in its related disciplines - health measurement, management etc.

Another important issue is that there are few jobs noted to come from government, with little commentary.  This is the area that one needs skilled personnel in order to effect the changes that are called for. Is this due to government jobs not being advertised in these online job portals?

There are numerous typos - like one fourth which is usually described as a quarter. I have identified these and put them in sticky notes in the attached manuscript. I have highlighted areas that need further clarity. I also think that some of the figures could be revised and taken out.

Author Response

(The authors gave the same response as above.)

Round 2

Reviewer 1 Report

Unfortunately these changes are not nearly sufficient to meet the requirements of "Major revisions". 

Furthermore, they have copied phrasing relating the Definition of a Public Health Job from "Public health job advertisements in Australia and New Zealand: a changing landscape" by Watts et al. It's only a few words, but given the extremely quick turn around for a "major revision" and the evidence of copying, I am not convinced that I need to read any further.